# REPRESENTATION MITOSIS IN WIDE NEURAL NETWORKS

## ABSTRACT

Deep neural networks (DNNs) defy the classical bias-variance trade-off: adding parameters to a DNN that interpolates its training data will typically improve its generalization performance. Explaining the mechanism behind this "benign overfitting" in deep networks remains an outstanding challenge. Here, we study the last hidden layer representations of various state-of-the-art convolutional neural networks and find evidence for an underlying mechanism that we call *representation mitosis*: if the last hidden representation is wide enough, its neurons tend to split into groups which carry identical information, and differ from each other only by a statistically independent noise. Like in a mitosis process, the number of such groups, or "clones", increases linearly with the width of the layer, but only if the width is above a critical value. We show that a key ingredient to activate mitosis is continuing the training process until the training error is zero.

## 1 INTRODUCTION

Deep neural networks (DNN) routinely have enough parameters to achieve zero training error, even with random labels (Zhang et al., 2017; Arpit et al., 2017). In defiance of the classical bias-variance trade-off, the performance of these *interpolating classifiers* continuously improves as the number of parameters increases well beyond the number of training samples (Geman et al., 1992; Neyshabur et al., 2015; Spigler et al., 2019; Nakkiran et al., 2020). Despite recent progress in describing the implicit bias of stochastic gradient descent towards "good" minima (Gunasekar et al., 2018a;b; Soudry et al., 2018; Ji & Telgarsky, 2019; Arora et al., 2019; Chizat & Bach, 2020), and the detailed analysis of solvable models of learning (Advani et al., 2020; Neal et al., 2018; Mei & Montanari, 2019; Belkin et al., 2019; Hastie et al., 2019; d'Ascoli et al., 2020; Adlam & Pennington, 2020; Lin & Dobriban, 2020; Geiger et al., 2020), the mechanisms underlying this "benign overfitting" (Bartlett et al., 2020) in *deep* NNs remain unclear, especially since "bad" local minima exist in their optimisation landscape and SGD can reach them (Liu et al., 2020).

In this paper, we describe a phenomenon in wide, deep neural networks that we call *representation mitosis* and which offers a possible mechanism for benign overfitting. We illustrate this mechanism in Fig. 1 for a family of increasingly wide DenseNet40s (Huang et al., 2017) on CIFAR10 (Krizhevsky et al., 2009). The blue line in Fig. 1 shows how the average classification error (error) approaches the performance of a large ensemble of networks ($\text{error}_\infty$) as the width of the network increases (Geiger et al., 2020). Consistently with Zagoruyko & Komodakis (2016), we find that their performance improves continuously with width. For simplicity we will refer to the width $W$ of last hidden representation as the width the network. When $W$ is greater than 350 a network becomes wide enough to reach zero training error (see Fig 9-c in Sec A.2) and the error decays approximately as $W^{-1/2}$. We make our key observation by performing the following experiment: we randomly select a number $w_c$ of neurons from the last hidden layer of the widest DenseNet40, and remove all the other neurons from that layer as well as their connections. We then evaluate the performance of this "chunk" of $w_c$ neurons, *without* retraining the network. The performance of chunks of varying sizes is shown in the same figure in orange. There are clearly two regimes: for small chunks, the error decays faster than $w_c^{-1/2}$, while beyond a critical chunk size $w_c^*$ (shaded area), the error of a chunk of $w_c$ neurons is roughly the same as the one of a full network with $w_c$ neurons. Furthermore, the error of the chunks decays with the same power-law $w_c^{-1/2}$ beyond this critical chunk size.

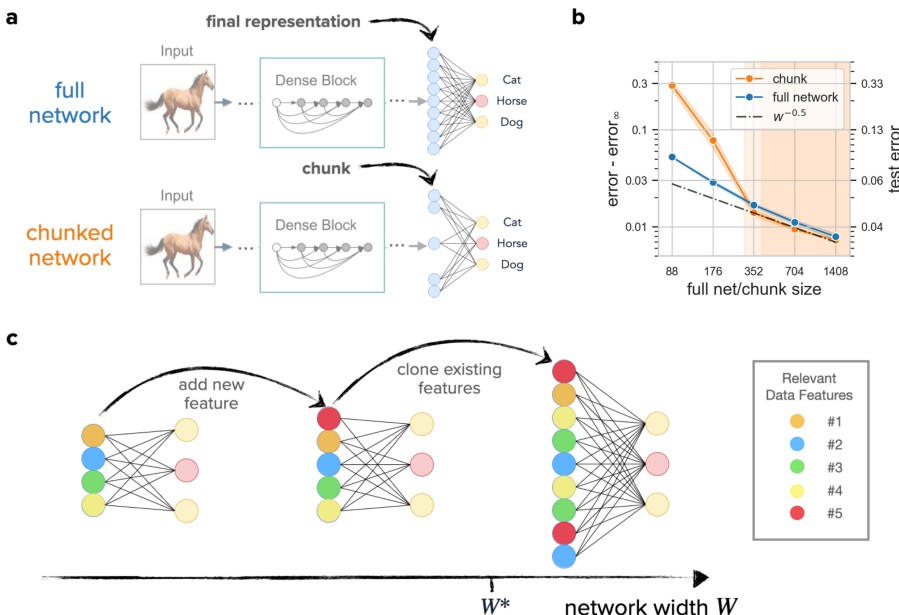

Figure 1: **The mechanism of representation mitosis.** **(a)** We analyse the final representations of deep neural networks (DNN), namely the activities of the last hidden layer of neurons (light blue) We focus on the performance and the statistical properties of randomly chosen subsets of $w_c$ neurons which we call "*chunks*". In the chunked network shown here, $w_c = 5$ out of 9 neurons are held, and used to predict the output. **(b)** As we increase the size of the chunk $w_c$ in a state-of-the-art DNN, here a DenseNet40, the test error of the chunked network (orange line) becomes similar to the test error of a full network of width $W = w_c$ (blue line). In this regime, which is reached when $w_c$ is larger than a threshold $w_c^*$ (shaded area) the error approaches its asymptotic value $\text{error}_\infty$ as a power-law $w_c^{-1/2}$ (dashed line). **(c)** Illustration of three final representations for networks of increasing width. As the width of the network increases, additional neurons fit to new features of the data (red neuron). As the network width goes beyond a critical width $W^*$, additional neurons instead copy features already learnt from data, and form what we call clones. This mechanism, which we call representation mitosis, is suggested by the $w_c^{-1/2}$ decay of the chunk error, and by the statistical analysis we present in this paper.

This observation suggests that the final hidden representation of an input in a trained, wide DNN is highly redundant beyond the critical width $w_c^*$. The decay rate of $-1/2$ in particular implies that in this regime chunks of $w_c$ neurons can be thought as statistically independent estimators of the same features of the data, differing only by a small, uncorrelated noise. This suggests a possible mechanism for benign overfitting: as the network becomes wider, additional neurons are first used to learn new features of the data. Beyond the critical width $w_c^*$, additional neurons in the final layer don't fit to new features in the data, and hence over-fit; instead, they make a copy, or a *clone*, of a feature that is already part of the final representation. The last layer thus splits into more and more clones as the networks grows wider in a process akin to *mitosis* in cell biology (Alberts et al., 2015), as we illustrate in the bottom of Fig. 1. The accuracy of these wide networks then improves with their width because the network implicitly averages over an increasing number of clones in its representations to make its prediction. We thus call this effect *representation mitosis*.

This paper provides a quantitative analysis of representation mitosis in non-trivial datasets and architectures. Our main findings can be summarised as follows:

1. A chunk of $w_c$ random neurons of the last hidden representation of a wide neural network predicts the output with an accuracy which scales with $w_c^{-1/2}$ if the layer is wide enough and $w_c$ is large enough. In this regime we call the chunk a "clone";

2. Clones can be linearly mapped one to another, or to the full representation, with an error which can be described as uncorrelated random noise.

3. Clones are created by training a well regularised model until the model reaches zero training error. If training is stopped too early (e.g. when the training accuracy is similar to the test accuracy), or if the training is performed without sufficient regularization, 1. and 2. do not take place, even if the last representation is very wide.

# 2 METHODS

## 2.1 NEURAL NETWORK ARCHITECTURES

We report experimental results obtained with various architectures (fully connected networks, Wide-ResNet-28, DenseNet40, ResNet50) and various data sets (CIFAR10/100 Krizhevsky et al. (2009), ImageNet Deng et al. (2009)). We trained all the networks with SGD with momentum and weight decay following the standard practice to set the relevant hyperparameters. We give detailed information on our training setups in A.1. All our experiments are run on Volta V100 GPUs. In the following, we discuss how we vary the architecture of the models.

**Fully-connected networks on MNIST**   We train a fully-connected network to classify the parity of the MNIST digits (LeCun & Cortes, 1998) (pMNIST) following the protocol of Geiger et al. (2020). MNIST digits are projected on the first ten principal components, which are then used as inputs of a five layer fully-connected network (FC5). The four hidden representations have the same width $W$ and the output is a real number whose sign is the predictor of the parity of the input digit.

**Wide-ResNet-28 and DenseNet40 on CIFAR10/100**   We train CIFAR10 and CIFAR100 on family of Wide-ResNet-28 (Zagoruyko & Komodakis, 2016) (WR28). The number of last hidden neurons in a WR28_$n$ is $64 \cdot n$, obtained after average pooling the last $64 \cdot n$ channels of the network. In our experiments we also analyze two narrow versions of the standard WR28_1 which are not typically used in the literature. We name them WR28_0.25 and WR28_0.5 since they have 1/4 and 1/2 of the number of channels of WR28_1. Our implementation of DenseNet40 follows the DeseNet40-BC variant (Huang et al., 2017). We vary the number of input channels $c$ in $\{8, 16, 32, 64, 128, 256\}$. The number of last hidden features for this architecture is $5.5 \cdot c$.

**ResNet50 on ImageNet**   We modify the ResNet50 architecture (He et al., 2016) multiplying by a constant factor $c \in \{0.25, 0.5, 1, 2, 4\}$ the number of channels of all the layers after the input stem. When $c = 2$ our networks differ from the standard Wide-ResNet50_2 since Zagoruyko & Komodakis (2016) only double the number of channels of the bottleneck of each ResNet block. As a consequence in our implementation the number of features $w$ after the last pooling layer is $w = 2048 \cdot c$ while in Zagoruyko & Komodakis (2016) $w$ is fixed to 2048.

## 2.2 ANALYSIS METHODS

**Reconstructing the wide representation from a smaller chunk**   To assess the predictive power of the chunk representations we search for the best linear map $\mathbf{A}$, of dimensions $W \times w$, able to minimise the squared difference $(\mathbf{x}^{(W)} - \hat{\mathbf{x}}^{(W)})^2$ between the $W$ activations of the full layer representation $(\mathbf{x}^{(W)})$ and the activations predicted from a chunk of size $w$,

$$\hat{\mathbf{x}}^{(W)} = \mathbf{A}\mathbf{x}^{(w)}. \tag{1}$$

This least squares problem is solved with ridge regression (Hastie et al., 2001) with regularization set to $10^{-8}$, and we use the $R^2$ coefficient of the fit to measure the predictive power of a given chunk size. The multi-output $R^2$ value is computed as an average of the W single-output $R^2$ values corresponding to the different coordinates, weighted by the variance of each coordinate. We further compute the covariance matrix $C_{ij}$ (of dimensions $W \times W$) of the residuals of this fit, and then obtain the correlation matrix as

$$\rho_{ij} = \frac{C_{ij}}{\sqrt{C_{ii}C_{jj}} + 10^{-8}}, \tag{2}$$

with a small regularisation in the denominator to avoid instabilities when the standard deviation of the residuals falls below machine precision. To quantify the independence of the chunk representations we take the average of the absolute values of the non-diagonal entries of the correlation matrix $\rho_{ij}$. For short we refer to this quantity as a 'mean correlation'.

**Reproducibility** We provide code to reproduce our experiments and our analysis online at https: //anonymous.4open.science/r/representation_mitosis-EB80.

# 3 RESULTS

**The test error of chunks of $w_c$ neurons of the final representation asymptotically scales as $w_c^{-1/2}$** The mechanism of representation mitosis is inspired by the following experiment: we compute the accuracy of models obtained by selecting a random subset of $w_c$ neurons from the final hidden representation of a wide neural network. We consider three different data sets (pMNIST, CIFAR10 and CIFAR100) and trained networks of width $W = 512$ for pMNIST and CIFAR10, and $W = 1024$ for CIFAR100. In all these cases, $W$ is large enough to be firmly in the regime where the accuracy of the networks scales (approximately) as $W^{-1/2}$ (see Fig. 2). We select $w_c$ neurons at random and we compute the test accuracy of a network in which we set to zero the activation of all the other $w - w_c$ neurons. Importantly, we do not fine-tune the weights after selecting the $w_c$ neurons: all the parameters are left unchanged, except that the activations of the "killed" neurons are not used to compute the output. We take 500 random samples of neurons for each chunk width $w_c$.

In Fig. 3 we plot the test error of the chunked models as a function of $w_c$ (orange lines). In all the three networks the behaviour is similar. When $w_c$ becomes larger than a critical value $w_c^*$, which depends on the dataset and architecture used, the test error decays as $w_c^{-1/2}$ with the chunk size, the same law observed for full networks of the

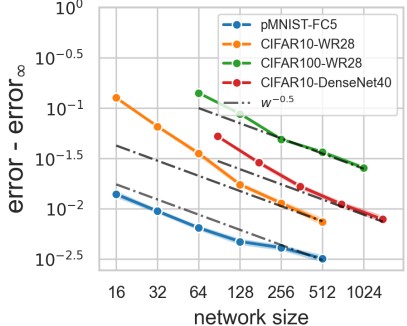

Figure 2: **Scaling of the test error with width for various DNN** The average test error of neural networks with various architectures approaches the test error of an ensemble of such networks as the network width increases. The network size shown here is the width of the final representation. For large width, we find a power-law behaviour error $-$ error$_\infty \propto W^{-1/2}$ across data sets and architectures. Full experimental details in Sec. 2.1

same width (Fig. 2). This implies that a model obtained by selecting a random chunk of $w_c > w_c^*$ neurons from a wide final representation behaves similarly to a full network of width $W = w_c$. Furthermore, a decay with rate $-1/2$ suggests that the final representation of the wide networks can be thought of as a collection of statistically independent estimates of a finite set of data features relevant for classification. Adding additional neurons to the chunk hence reduces their prediction error in the same way an additional measurement reduces the measurement uncertainty, leading to the $-1/2$ decay.

At smaller $w_c < w_c^*$ instead, the test error of the chunked models decays faster than $w_c^{-1/2}$ in all the cases we considered, including the DenseNet architecture trained on CIFAR10 shown in Fig. 1. In this regime, adding neurons to the the final representation, improves the quality of the model significantly quicker than it would in independently trained models of the same width (see Fig. 1 for a pictorial representation of this process). We call chunks of neurons of size $w_c \geq w_c^*$ *clones*. In a wide network, clones can exist only if the width $W$ of the last representation is larger than $w_c^*$, and the maximum number of clones is $W/w_c^*$. In the following we characterize more precisely the properties of the clones.

**Clones have the same expressive power of the full representation** A well trained deep network often represents the salient features of the data set well enough to achieve (close to) zero classification error on the training data. In the top panels of Fig. 4, we show that wide networks are able to interpolate their training set also using just a subset of $w_c > w_c^*$ random neurons: the dark orange profiles show that when the size of a chunk is greater than $\sim 50$ for pMNINST, 100 for CIFAR10 and

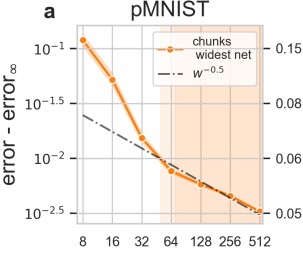 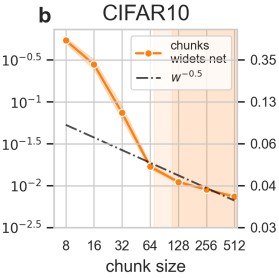 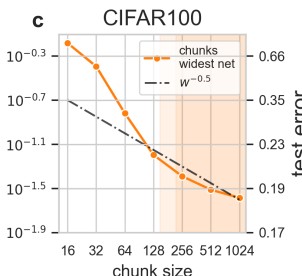

Figure 3: **Scaling of the test error of chunks of neurons extracted from the final representation of wide NNs** We plot how the test error of chunked networks approaches the error of an ensemble of chunked networks as the chunk size $w_c$ increases. Chunks are formed by selection a number of $w_c$ neurons at random from the final representation of the widest networks: a FC5 on pMNIST (width $W = 512$), and Wide-ResNet-28 for CIFAR10 ($W = 512$) and CIFAR100 ($W = 1024$). The shaded regions indicate regions where the error of the chunks with $w_c$ neurons decays as $w_c^{-1/2}$.

200 for CIFAR100, the predictive accuracy on the training set remains almost 100%. Beyond $w_c^*$, the neurons of the final representation therefore become redundant, since the training error remains (close to) zero even after removing neurons from it. We call a chunk of neurons a clone if it fully captures the relevant features of the data, up to some uncorrelated random noise.

**Clones reconstruct almost perfectly the full representation** From a geometrical perspective, the important features of the final representation correspond to directions in which the data landscape show large variations (Bengio et al., 2013). A clone can be seen as a chunk that is wide enough to encode almost exactly these directions, but using much less neurons than the full final representation. We analyze this aspect by fitting all the $W$ activations starting from a random chunk of $w_c$ activations with ridge regression with a small regularization penalty according to Eq. (1). The blue profiles in Fig. 4, bottom panels, show the $R^2$ coefficient of fit as a function of the chunk size $w_c$ for pMNIST (left), CIFAR10 (center), CIFAR100 (right). When $w_c$ is really small, say below 6 for pMNIST, 20 for CIFAR10 and 60 for CIFAR100, the $R^2$ coefficient grows almost linearly with $w_c$. In this regime, adding a randomly chosen activation from the full representation to the chunk increases substantially $R^2$. When $w_c$ becomes larger $R^2$ reaches almost one and the representation enters what we call a mitosis phase. This transition happens when $w_c$ is still much smaller than $W$ and correspond approximately to the regime in which the test error starts scaling with the inverse square root of $w_c$ (see Fig. 3). The almost perfect reconstruction of the original data landscape with few neurons can be seen as a consequence of the low *intrinsic dimension* of the representation (Ansuini et al., 2019). The ID of the widest representations gives a lower bound on the number of coordinates required to describe the data manifold, and hence on the neurons that a chunk needs in order to have the same classification accuracy as the whole representation. The ID of the last hidden representation is 2 in pMNIST, 12 in CIFAR10, 14 in CIFAR100, numbers which are much lower that the width at which a chunk can be considered a clone.

**Clones differ from each other by uncorrelated random noise** In the mitosis regime, the small residual difference between the representation chunks and the full representation can be approximately described as statistically independent random noise. The green profile of Fig. 4, bottom panels, show the mean absolute non-diagonal correlation of the residuals of the linear fit, a measure which indicates the level of correlation of the chunk representations and the full representation (see Methods). Before the mitosis width $w_c^*$, the residuals are not only large, but also significantly correlated, but as their width increases above $w_c^*$ the correlation drops basically to zero. Therefore, in network which are wider than $w_c^*$ any two chunks of equal size $w_c > w_c^*$ can be effectively considered as equivalent copies, or clones, of the same representation, differing only by a small and non-correlated noise, consistently with the scaling law of the error shown in Fig. 3.

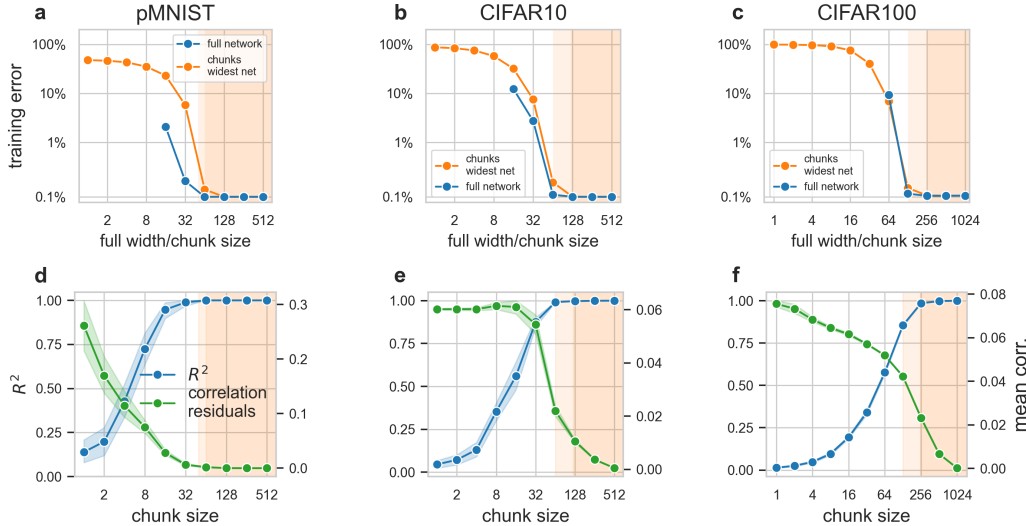

Figure 4: **The three signatures of representation mitosis (i)** The training errors of the full networks (blue) and of the chunks taken from the widest network (orange) approach zero beyond a critical width / chunk size, resp. (panels a-c). **(ii)** The final representation of the widest network can be reconstructed from a chunk using linear regression (1) with an explained variance $R^2$ close to 1 (blue lines in panels d-f). **(iii)** The residuals of the linear map can be modeled as independent noise: we show this by plotting the mean correlation of these residuals (green line, panels d-f), averaged over 100 reconstructions starting from different chunks. A low correlation at high $R^2$ indicates that the chunk contains the information of the full representation with some statistically independent noise. *Experimental setup:* FC5 on pMNIST, Wide ResNet-28 on CIFAR10/100. Full details in Methods section 2.1

**The dynamics of mitosis** In the previous paragraphs we set forth evidence in support of the hypothesis that large chunks of the final representation of wide DNNs behave approximately like an ensemble of independent measures of the full feature space. This allowed us to interpret the decay of the test error of the full networks with the network width observed empirically in Fig. 2. The three conditions that a chunked model satisfies in the regime in which its test error decays as $w_c^{-1/2}$ are represented in Fig. 4: (i) the training error of the chunked model is close to zero; (ii) the chunked model can be used to reconstruct the full final representation with an $R^2 \sim 1$ and (iii) the residuals of this reconstruction can be modeled as independent random noise. These three conditions are all observed at the end of the training. We now analyze the *dynamics* of mitosis. We will see that to enter the mitosis regime, models not only need to be wide enough, but also, crucially, they need to be trained to maximise their performance.

Clones are formed in two stages, which occur at different times during training. The first phase begins as soon as training starts: the network gradually adjusts the chunk representations in order to produce independent copies of the data manifold. This can be clearly observed in the left panel of Fig. 5, which depicts the mean correlation between the residuals of the linear fit from the chunked to the full final representations of the network, the same quantity that we analyze in Fig. 4, but now as a function of the training epoch. Both Figs. 4 and 5 analyse the WR28-8 on CIFAR10. As training proceeds, the correlations between residuals diminish gradually until epoch 160, and becomes particularly low for chunks greater than 64. After epoch 160 further training does not bring any sizeable reduction in their correlation. At epoch 160 the full network achieves zero error on the training set, as shown in orange in the middle and right panels of Fig. 5. This event marks the end of the first phase, and the beginning of the second phase of the mitosis process where the training error of the clones keeps decreasing while the full representation (blue) has already reached zero training error. For example, chunks of size 64 at epoch 150 have training errors comparable to the test error (dashed line of the

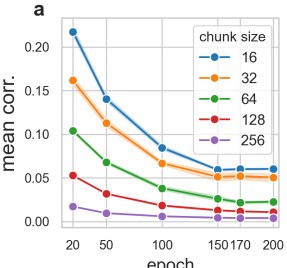 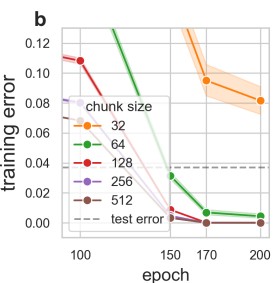 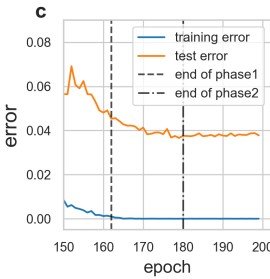

Figure 5: **The onset of mitosis during training. a:** As in Fig. 4, we show the mean correlation of the residuals of the linear reconstruction of the final representation from chunks, but this time as a function of training epochs. A small correlation indicates that the reconstruction error in going from chunks to final representation can be modelled as independent noise. Data obtained from the same WR28_8 trained on CIFAR10 as in Fig. 4. **b:** Training error during training for chunks of different sizes. After the network has reached zero training error at $\sim 160$ epochs, continuing to train improves the training accuracy of the chunks. **c:** Test and training error during training for the full network. Between epoch 160 and 180 the clones of the full network progressively achieve zero training error. In the same epochs, one observes a small improvement in the test error.

middle panel). In the subsequent $\sim 20$ epochs the training error of clones of size 128 and 256 reaches exactly zero, and the training error of chunks of size 64 reaches a plateau.

Importantly, both phases of mitosis improve the generalization properties of the network. This can be seen in the right panel of Fig. 5, which reports training and test error of the network, with the two mitosis phases highlighted. The figure shows that both mitosis phases lead to a reduction in the test error, although the first phase leads by far to the greatest reduction, consistently with the fact that the greatest improvements in accuracy typically arise during the first epochs of training. The mitosis process can be considered finished around epoch 180, when all the clones have reached almost zero error on the training set. After epoch 180 we also observe that the test error stops improving. In Sec. A.2 we report the same analysis done on CIFAR100 (see Fig. 7) and CIFAR10 trained on a DenseNet40 (see Fig. 9-(d-e-f)).

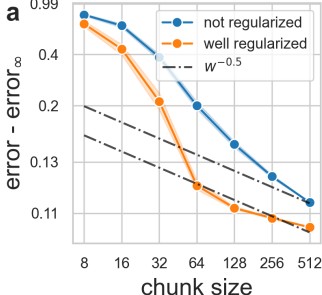 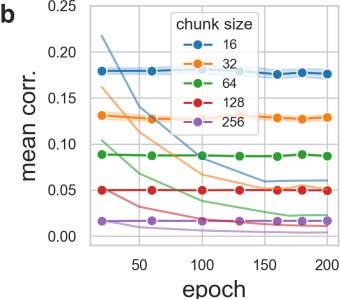

Figure 6: **A network trained without regularization on CIFAR10. a:** the test error of chunks of a Wide-ResNet28-8 trained without data augmentation and weight decay (blue) and for a well-regularized network (orange, taken from Figure 3-b). **b:** Mean correlation between residuals of the linear reconstruction of the full representation from chunks of different sizes for two networks: one trained without data augmentation and weight decay (thick lines), and one using state-of-the-art techniques (thin lines, same data as in Fig. 5-a).

### 3.1 LIMITATIONS

**Clones appear only in well-trained regularized networks**   So far in this work we have shown only examples of networks and datasets in which representation mitosis takes place. However, the onset of this mechanism is related to nontrivial details in the learning algorithm. If the network is not trained using state-of-the-art procedures, none of the signatures described above appear even if the width of the final representation is much larger than $W^*$. Figure 6 shows the case of the Wide-ResNet28-8 analyzed in Fig. 5 trained on exactly the same dataset (CIFAR10) but without data augmentation and weight decay. As shown in panel **a**, in the network trained without sufficient regularization (blue line) the error does not scale as $w_c^{-1/2}$, not even for large $w_c$. This, as we have seen, indicates that the last representation cannot be split in chunks. Correspondingly, the chunks cannot be mapped to the full representation by a linear transformation: the mean correlation of the residuals of the linear map of the chunks to the full representation remains approximately constant during training, and is always much higher then what we observed for the same architecture and dataset, when training is performed with weight decay and data augmentation (panel **b**). We performed a similar analysis on the DenseNet40 (see Fig. 10), observing an analogous trend.

**Reaching mitosis on ImageNet**   A second scenario where we didn't observe the hallmarks of mitosis is ImageNet. We trained a family of ResNet50 where we multiply all the channels of the layers after the input stem by a constant factor $c \in \{0.25, 0.5, 1, 2, 4\}$. In this manner the widest final representation we consider consists of 8192 neurons, which is four times wider than both the standard ResNet50 (He et al., 2016) and its wider version (Zagoruyko & Komodakis, 2016) (see Sec. 2.1). We trained all the networks following the standard protocols and achieved test errors comparable or slightly lower than those reported in the literature (see Sec. A.1).

In this setting none of the elements associated with a development of independent clones can be observed. The scaling of the test error of the chunks is steeper than $w_c^{-1/2}$ (see Fig. 8-a) suggesting that chunks remain significantly correlated to each other. Figure 8-b shows that the mean correlation of the residuals does not decrease during training, as it happens for the networks we trained on CIFAR10 and CIFAR100. We conclude that a representation with 8192 neurons seems too narrow to encode all the relevant features redundantly on ImageNet, which has 1000 classes. Indeed, the top-1 classification error of the largest ResNet50 on the training set is $\sim 92\%$, well below the interpolation threshold and a chunk as large as 4096 activations is not able to reconstruct all the relevant variations of the data as it does in the cases analyzed in Sec. 3 (see Fig. 8-c).

## 4 DISCUSSION

This work is an attempt to explain the apparently paradoxical observation that over-parameterization boosts the performance of DNNs. This "paradox" is actually not a peculiarity of DNNs: if one trains a prediction model with $n$ parameters using the same training set, but starting from independent initial weights and receiving samples in an independent way, one can obtain, say, $m$ models which, in suitable conditions, provide predictions of the same quantity with independent noise due to initialization, SGD schedule, etc. If one estimates the target quantity by an ensemble average, the statistical error will (ideally) scale with $m^{-1/2}$, and therefore with $N^{-1/2}$, where $N = n\,m$ is the total number of parameters of the combined model. This will happen even if $N$ is much larger than the number of data.

What is less trivial is that a DNN is able to accomplish this scaling within a single model, in which all the parameters are optimised collectively via minimization of a single loss function. Our work describes a possible mechanism at the basis of this phenomenon in the special case of a neural networks in which the last layer is very wide. We observe that if the layer is wide enough, random subsets of its neurons can be viewed as approximately independent representations of the same data manifold (or clones). This implies a scaling of the error with the width of the layer as $W^{-1/2}$, which is qualitatively consistent with our observations.

**The impact of network architecture.**   The capability of a network to produce statistically independent clones in its last layer is architecture-dependent, since we find that the network width $W$ required to enter the mitosis regime needs to be large, as in Fig. 3. The total number of parameters

in a DNN scales with $W$ in a manner that, in general, depends on the architecture. If the mitosis mechanism we propose is correct, the scaling of the test error with $N$ will thus be, in general, architecture-dependent. At the same time, we also verified that increasing the width of *only* the final representation is not sufficient to enter the mitosis regime. We give an example of this effect in Fig. 11 in Sec. A.2, where we show that the test error of a Wide-ResNet28 on CIFAR10 does not decrease if only the width of the final representation is increased, while the rest of the architecture is kept at constant width.

**The impact of training.**   Even for wide enough architectures, clones appear only if the training schedule is appropriately chosen. In our examples, by stopping the training too early, for example when the training error is similar to the test error, the chunks of the last representation would not become entirely independent from one another, and therefore they could not be considered clones. In fact, we have seen that the separation of the clones is completed only when the test error on a model restricted to each clone becomes very small.

**Neural scaling laws**   Capturing the asymptotic performance of neural network via scaling laws is another active research area. Hestness et al. (2017) gave an experimental analysis of scaling laws w.r.t the training data set size in a variety of domains. Rosenfeld et al. (2020); Kaplan et al. (2020) experimentally explored the scaling of the generalisation error of deep networks with the number of parameters/data points across architectures and application domains for supervised learning, while Henighan et al. (2020) identified empirical scaling laws in generative models. Geiger et al. (2020) found that the generalisation error of fully-connected networks trained on the pMNIST task we also consider scales with the number of parameters. Bahri et al. (2021) showed the existence of four scaling regimes and described them theoretically in the setting of random features, which is an instance of lazy learning (Chizat et al., 2019). Sharma & Kaplan (2020) relate the exponent of scaling laws with respect to the number of training samples to the dimension of the data manifold. Our analysis supports the hypothesis that the test error in wide neural networks scales as the inverse square root of the width, if the width is large enough, and if the classification task is not too complex.

**Relation to theoretical results in the mean-field regime.**   Our empirical results also agree with recent theoretical results that were obtained for two-layer neural networks (Mei et al., 2018; Rotskoff & Vanden-Eijnden, 2018; Chizat & Bach, 2018; Sirignano & Spiliopoulos, 2019; Goldt et al., 2019; Refinetti et al., 2021). These works characterise the optimal solutions of two-layer networks trained on synthetic datasets with some controlled features. In the limit of infinite training data, these optimal solutions correspond to networks where neurons in the hidden layer duplicate the key features of the data. These "denoising solutions" or "distributional fixed points" were found for networks with wide hidden layer (Mei et al., 2018; Rotskoff & Vanden-Eijnden, 2018; Chizat & Bach, 2018; Sirignano & Spiliopoulos, 2019) and wide input dimension (Goldt et al., 2019; Refinetti et al., 2021). Another point of connection with the theoretical literature is the concept of *dropout stability*. A network is said to be $\epsilon$-dropout stable it its training loss changes by less than $\epsilon$ when half the neurons are removed at random from each of its layers Kuditipudi et al. (2019). Dropout stability has been rigorously linked to several phenomena in neural networks, such as the connectedness of the minima of their training landscape Shevchenko & Mondelli (2020); Nguyen et al. (2021).

**Implicit ensembling in deep learning**   The success of various deep learning architectures and techniques has been linked to some form of ensembling. Veit et al. (2016) proposed that the performance of ResNets (He et al., 2016) stems from an effective ensembling of shallower networks due to the residual connections. The successful **dropout** regularisation technique (Hinton et al., 2012; Srivastava et al., 2014) samples from an exponential number of "thinned" networks during training to prevent co-adaptation of hidden units. While this can be seen as a form of (implicit) ensembling, here we make the observation that co-adaptation of hidden units in the form of clones occurs *without* dropout, and is crucial for their improving performance with width. Recent theoretical work on random features (d'Ascoli et al., 2020; Adlam & Pennington, 2020; Lin & Dobriban, 2020) also suggests that ensembling and over-parameterisation are two sides of the same coin, at least in the lazy regime of neural networks (Chizat et al., 2019), where the weights of the networks' hidden layers don't move much during training. In this paper, instead we focus on networks in the feature learning regime where weights move significantly. We hope that our results motivate theoretical studies along those lines in the feature learning regime.

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

# A APPENDIX

## A.1 HYPERPARAMETERS USED AND TRAINING PROCEDURES

**Fully-connected networks on MNIST** We train the fully-connected networks for 5000 epochs with stochastic gradient descent using the following hyperparameters: batch size = 256, momentum = 0.9, learning rate = $10^{-3}$, weight decay = $10^{-2}$. We decrease the initial learning rate with a cosine schedule.

**Wide-ResNet-28 and DenseNet40-BC on CIFAR10/100** All the models are trained for 200 epochs with stochastic gradient descent with a batch size = 128, momentum = 0.9, and cosine annealing scheduler starting with a learning rate of 0.1. The training set is augmented with horizontal flips with 50% probability and random cropping the images padded with four pixels on each side. On CIFAR10 trained on WR28 we select a weight decay equal to $5 \cdot 10^{-4}$ and label smoothing magnitudes equal to 0.1 for WR28_{0.25, 0.5, 1, 2}. On CIFAR10 trained on Densenet40-BC we set select a weight decay equal to $5 \cdot 10^{-4}$ and label smoothing magnitudes equal to 0.05 for all the networks On CIFAR100 trained on WR28 we set weight decays equal to $\{10, 7, 5, 5, 5\} \cdot 10^{-4}$ and label smoothing magnitudes equal to $\{0.1, 0.07, 0.05, 0, 0\}$ for WR28_{1, 2, 4, 8, 16} respectively. All the hyperparameters were selected with a small grid search.

**ResNet50 on ImageNet** We train all the ResNet50 with mixed precision (Micikevicius et al., 2018) for 120 epochs with a weight decay of $4 \cdot 10^{-5}$ and label smoothing rate of 0.1 (Bello et al., 2021). The input size is $224 \times 224$ and the training set is augmented with random crops and horizontal flips with 50% probability. The per-GPU batch size is set to 128 and is halved for the widest networks to fit in the GPU memory. The networks are trained on 8 or 16 Volta V100 GPUs so as to keep the batch size $B$ equal to 1024. The learning rate is increased linearly from 0 to $0.1 \cdot B/256$ (Goyal et al., 2017) for the first five epochs and then annealed to zero with a cosine schedule.

Table 1: Test accuracy (average over four runs)

| CIFAR10 | | CIFAR100 | | ImageNet (top1) | |
|---|---|---|---|---|---|
| network | accuracy | network | accuracy | network | accuracy |
| Wide-RN28_0.25 | 84.1 | Wide-RN28_1 | 70.4 | RN50_0.25 | 67.0 |
| Wide-RN28_0.5 | 90.3 | Wide-RN28_2 | 75.7 | RN50_0.5 | 74.1 |
| Wide-RN28_1 | 93.4 | Wide-RN28_4 | 79.6 | RN50_1 | 77.6 |
| Wide-RN28_2 | 95.2 | Wide-RN28_8 | 80.8 | RN50_2 | 79.1 |
| Wide-RN28_4 | 95.9 | Wide-RN28_16 | 81.9 | RN50_4 | 79.5 |
| Wide-RN28_8 | 96.1 | | | | |
| DenseNet40-BC (k=8) | 91.6 | | | | |
| DenseNet40-BC (k=16) | 93.9 | | | | |
| DenseNet40-BC (k=32) | 95.1 | | | | |
| DenseNet40-BC (k=64) | 95.7 | | | | |
| DenseNet40-BC (k=128) | 96.0 | | | | |

## A.2 ADDITIONAL EXPERIMENTS

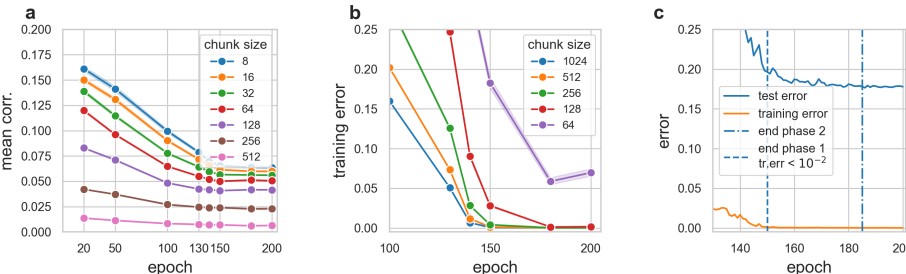

Figure 7: **Dynamics of mitosis on CIFAR100 a:**As in Fig. 4, we show the mean correlation of the residuals of the linear reconstruction of the final representation of a Wide-Resnet28_8 from chunks, but this time as a function of training epochs. A small correlation indicates that the reconstruction error in going from chunks to final representation can be modelled as independent noise. **b:** Training error of chunks of a Wide-Resnet28_8 and its full layer representation. From epoch 150 to epoch 185 the training error of the chunks with size 128/256 decreases below 0.5%, while for smaller chunks sizes it remains above 5%. Random chunks with size larger than 128/256 can fit the training set, thus having the same representational power as the whole network on the training data. For W > 128/256 the test accuracy is decaying approximately with the same law as that of independent networks with the same width (see Fig. 3). This picture suggests that for CIFAR100 the size of a clone is 128/256, slightly larger than the size of the clones in CIFAR10. **c:** Training and test error dynamics for the same Wide-ResNet28_8. After epoch 150 the training error of the full network remains consistently smaller than 0.1% (orange profile) while the test error continues to decrease until epoch 185 from 0.194 to 0.1765 (blue profile). In the same range of epochs (150-185) the training error of smaller chunks decreases sensibly (see panel **b**).

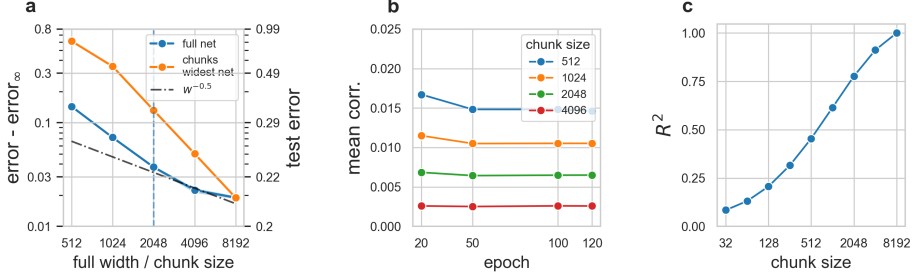

Figure 8: **No representation mitosis in ResNet50 on ImageNet a:** Decay of the test error as a function of the network width (blue) and for chunks of the widest ResNet50 (orange) to the error of an ensemble of ResNet50_4. The ensemble consists of four networks. **b:** Mean correlation (see Sec. 2.2) of the residuals of the linear map of a chunk of the last hidden representation to the full representation. The network examined is ResNet50_4. **c:** $R^2$ coefficient of the ridge regression fit of a chunk of the last hidden representation of a ResNet50_4 to its full layer representation.

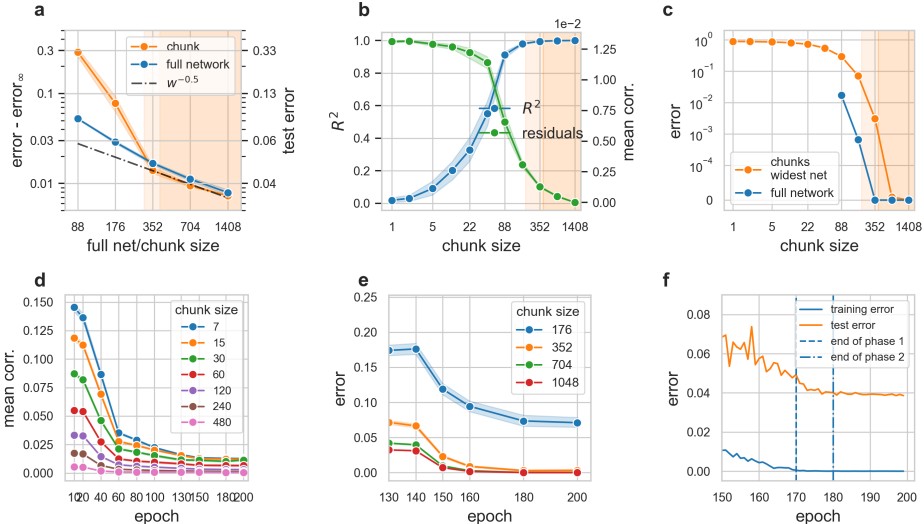

Figure 9: **The representation mitosis for a DenseNet40 architecture. a:** Decay of the test error of independent networks (blue) and chunks of the widest network (orange) to the error of an ensemble average of ten of the widest networks (DenseNet40-BC, k=128) **b:** Blue profile: $R^2$ coefficient of the ridge regression of a chunk of $w_c$ neurons ($x$-axis) to the full layer representation. Green profile: mean correlation of the residuals of the mapping as described in Sec. 2.2. **c:** Training error of various DenseNet40 of increasing width (blue) and of chunks of the widest architecture (orange). **d:** The mean correlation of the residuals from the linear reconstruction of the final representation from chunks of a given size for a DenseNet40-BC (k=128) during training. **e:** Training error dynamics of chunks of a DenseNet40-BC (k=128). **f:** Training and test error dynamics for a DenseNet40-BC (k=128).

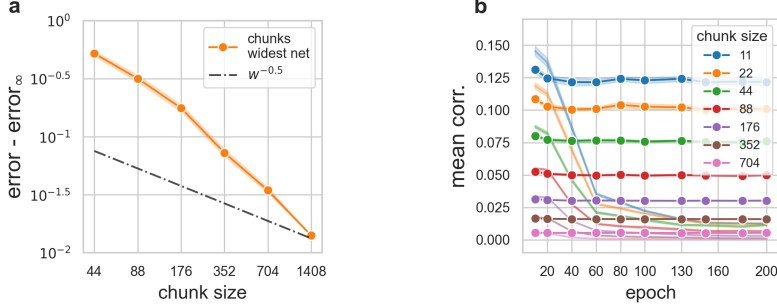

Figure 10: **A Densenet40 without mitosis** A DenseNet40-BC (k=128) trained on CIFAR10 without weight decay and data augmentation. This experiment reproduces on a DenseNet the analysis shown on a Wide-ResNet28 in Sec. 3.1. It shows that **a**: also in a DenseNet architecture not well regularized error -error$_\infty$ decays faster than $w_c^{-1/2}$ and **b**: the mean correlation of the residuals do not decrease during training. The thin profiles of panel **b** are the same as those shown in Fig. 9-d.

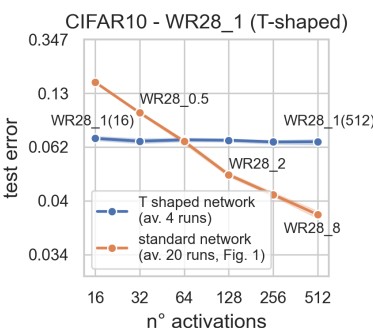

Figure 11: **Wide-RN28_1 with a wide output layer trained on CIFAR10** We tested whether it is enough to increase the width of the final representation to see mitosis, or if instead one has to increase the width of the full network. We trained a ResNet28_1 increasing only the number of channels in the last layer. We modified the number of output channels of the last block of conv4 and analysed the representation after average pooling, as we did in the other experiments. The network was trained for 200 epochs using the same hyperparameters and protocol described in Sec. 2. The figure shows that the test error of the modified ResNet28_1 for is approximately constant (blue profile). On the contrary when we increase the width of the whole network the test error decays to the asymptotic test error with an approximate scaling of $1/\sqrt{w}$ (orange profile). Therefore mitosis occurs when the width of the whole architecture is increased, rather than just the width of the final layer.

