# OpenReview forum: "Representation mitosis in wide neural networks"
_ICLR.cc/2022/Conference — ICLR 2022 Submitted_

### Official Review · Reviewer_YoAs · 2021-10-27

**Correctness:** 3
**Technical Novelty And Significance:** 1
**Empirical Novelty And Significance:** 3
**Recommendation:** 6
**Confidence:** 4

**Main Review:**

The paper is mostly empirical in nature, and I think the authors performed a very nice set of carefully designed experiments to explore their novel "cloning" effect. I appreciate the variety of architectures and vision datasets considered, and I further appreciate the candor in Section 3.1 on Limitations. I also think that the analysis of reconstructing the wide representation from the smaller subset is a nice way of making the point about features being cloned and the information being present in the smaller subset.

I think two primary weaknesses involve the aforementioned limitations.

The first weakness is the necessity of using heavy regularization and complicated training procedures. For typical discussions of overparameterized networks -- e.g. ones that discuss the kernel learning regime -- regularization isn't necessary to see the double descent phenomenon highlighted in the Belkin et al. paper cited by the authors. In fact, gradient descent is supposed to automatically pick out the minimal norm solution, and such models overparameterized trained this way do not overfit even if they are not explicitly regularized. Thus, I conclude that the "cloning" effect observed by these authors cannot be the main explanation for the way DNNs appear to defy the classical bias-variance trade-off. Further, I wonder whether "cloning" is somehow an artifact of the augmentation+regularization used. Perhaps the authors could explore this more and/or come up with ways to disentangle these effects or explore cloning/mitosis in other overparameterized models so that its origin can be better understood?

Next, I think it's problematic that the cloning effect was not observed for ImageNet, though this is definitely a lesser weakness: it could potentially be explained by the fact that the authors were unable to get a wide enough final hidden-layer representation in comparison to the number of classes in ImageNet. As further evidence, the authors highlight that they were not able to train their widened ResNet50 to convergence, which was a prerequisite to see "cloning" in the other instances. However, I am worried -- since there isn't any strong theoretical explanation or intuition provided -- that the cloning observed is potentially an artifact of modeling simpler datasets -- e.g. ones where PCA would give reasonable representations -- and thus is not a very general phenomenon in deep neural networks.

Some additional comments:
- I would enjoy further discussion about what determines $w_c$ for a particular set of architecture, learning algorithm, and dataset. Does varying the depth of the network have any effect on this threshold? (In recent analyses of wide networks, the network's depth to width ratio plays an important role in determining when a network becomes "wide," so this seems like an appropriate scale to compare the width to.)
- The authors claim that the residual after reconstruction is simply white noise. Is it possible that higher order correlations than the covariance are present and not captured by the subset chunk?
- In the final paragraph, the authors say that they focus on networks in the feature learning regime, but I thought that to observe their mitosis effect that networks need to be sufficiently wide. Thus, I find their claim a little hard to believe. Do they think that there's another transition for even wider networks that are lazy learners, in which there is no cloning? (By the way, I think that in many ways, the infinite-width limit provides a natural starting point to explain mitosis, since the hidden representations in such networks are all independent at initialization and throughout training.)


**Summary Of The Paper:**

This paper investigates the overparameterization problem in deep learning through a set of experiments that study the final-layer hidden representations of wide and deep networks. The authors find that -- for wide enough networks and after training long enough -- that they can achieve similar test error from a model made by sampling only a subset of the final-layer hidden representations and then deleting the other connections in that layer. Furthermore, in such situations they can find a linear map from the subset that predicts the full set of final-layer representations, suggesting that the information contained in the deleted features is "cloned" and present in remaining subset. Overall, this effect and its dynamics are explored empirically across a number of models, datasets, and learning scenarios. Curiously, the cloning only appears in networks where data augmentation and weight decay is used, and the authors were unable to exhibit this effect on ImageNet.

**Summary Of The Review:**

I think this is a nice paper that demonstrates an interesting empirical effect and proposes a promising connection to the behavior of overparameterized deep neural networks. However, in its current form, I think the submission is marginally below the acceptance threshold. In particular, I am bothered by the fact that the effect appears only after training "using state-of-the-art procedures" and is absent for complicated datasets (the example in the paper being ImageNet).

Overall, I think this paper makes a nice set of observations and performs careful scientific experiments. With a little more clarity in terms of how general this effect is, I would be happy to accept it to ICLR.

### After Author Responses
I appreciate the comments of the authors. I still am concerned about how general the mitosis effect is, since it doesn't occur in simpler models. However, with the additional clarity, I have increased my score to be at the acceptance level.

---

> ### Author Response · Authors · 2021-11-15
> **Response to reviewer YoAs**
>
> We are grateful for the for the constructive comments provided. We try to address all the main concerns below:
>
> > *1 Heavy regularization, complicated training*
>
> The training protocols we used are neither complicated nor heavily regularized. We apologize if we have been misleading on this point: our procedure simply follows the most common protocols described in the literature (e.g. original Wide-ResNet paper https://arxiv.org/pdf/1605.07146.pdf, original Densenet paper https://arxiv.org/abs/1608.06993, cosine annealing scheduler https://arxiv.org/pdf/1608.03983v5.pdf).
> Our training scheme follows that of the original papers with small modifications to stay compliant with the recent training practices. On the widest networks the weight decay used (0.0005) is the same as most of the paper cited above (and many others), including the original paper on the Wide-ResNets.
>
> > *2 Generalization without regularization*
>
> This is a good point and a similar concern was also raised by Rev KFV6.
> It is true that overparameterized networks generalize also without regularization. Yet, the accuracy they reach is very far from the state-of-the-art performances that motivate the use of DNNs in the  practice. The main scope of this work was understanding better state-of-the-art models used by the community, not just simple settings or toy models.
>
> To be more quantitative let’s focus on the comparison shown in Fig. 6. The test error of the model trained without regularization (and that does not show mitosis) is 7%. When the model is trained with the right amount of regularization (and shows mitosis) the test error drops to 4%. We will modify the sentences (see e.g. the one reported by Rev. KFV6)  that have been misleading on this aspect.
>
> > *3 Disentangle effect of mitosis*
>
> We performed an additional test since the lack of clarity on this point was problematic also to other reviewers.
> Mitosis occurs if:
> 1) the model achieves zero error on the training set;
> 2) the model is trained with an optimal weight decay rate.
>
> See our reply to Rev vhrZ that expands more on this point.
>
> > *4 Imagenet and PCA*
>
> PCA provides good representation on MNIST. However we doubt that on CIFAR10/100, PCA or other “non-learned” representations could be even vaguely comparable to those produced by a convolutional network.
>
> The reason why we were not able to observe mitosis on ImageNet is because a ResNet50 is not expressive enough to interpolate the training set and therefore condition 1) above does not hold (as also noticed in Belkin et al).
> This difficulty could be overcome by making ResNet50 even larger (we agree on this point with the reviewer), or making the network deeper. We think that increasing the depth is the most promising direction for future works because:
>
> 1) the expressivity of a deep network increases with depth more than it does with width;
> 2) from a preliminary analysis on DenseNet100 we saw that the clone size decreases if the network is made deeper (see also rev vhrZ).
>
> However, training e.g. a Wide-ResNet152 on ImageNet is at present too costly for us. But we are firmly convinced that also on ImageNet one can observe representation mitosis provided conditions 1) and 2) are satisfied.
>
> > *5 Role of depth*
>
> A similar question was asked by rev vhrZ. We kindly ask to check our first answer of vhrZ **"minor comments"** (DenseNet100 vs Densenet40).
>
> > *6 Higher moments*
>
> It is true that it would be possible to have high order correlations even when first order correlations are zero. However, higher correlations would affect in some way also the measured linear correlations, apart from pathological cases of high symmetry which are unlikely to emerge spontaneously during training.
>
> > *7 Lazy regime*
>
> To test quantitatively if our networks operate on a lazy regime we analyzed the stability of the activations as proposed in Chizat et al. (https://arxiv.org/pdf/1812.07956.pdf). We extract the activations before the last ReLU layer of the widest DenseNet40 studied in the paper. We compare the ratio  $r$ of the test set activations that DON’T change sign from epoch 1 to epoch 200. In Chizat et al, lazy training is associated with a $r$ close to 1; in our case this $r$ is 0.514 regime in which the network is far from being ‘lazy’ (see Chizat et al.).
> The range of widths explored in our paper spans almost two orders of magnitude going from ResNet28_1 to Wide-ResNet28_16/DenseNet40-BC128. Our purpose was studying state-of-the-art networks used in practice. These architectures do not operate on the lazy regime even when they are made wide (at least on the ranges we analysed).
>
> **Conclusion**
> We thank the reviewer for acknowledging that our paper “makes a nice set of observations and performs careful scientific experiments”. We hope that our additional explanation can alleviate your concerns regarding the clarity of our results and help to raise the score. We’ll be happy to reply to any further questions.

---

> ### Author Response · Authors · 2021-11-29
> **Thank you for increasing your score!**
>
> In the final revision we will be more explicit about the necessary conditions to observe mitosis, i.e.:
>
> 1) the model achieves zero error on the training set;
> 2) the model is trained with an optimal weight decay parameter.
>
> which we think are fairly general and often occur in practice. We will clarify better what it missing when mitosis does not show up and how to obtain it (i.e. deeper/larger models when condition 1) is missing or adjusting the regularization when 2) is missing).
>
> Thank you again for revising your score!

---

### Official Review · Reviewer_7hrx · 2021-11-03

**Correctness:** 3
**Technical Novelty And Significance:** 2
**Empirical Novelty And Significance:** 2
**Recommendation:** 5
**Confidence:** 4

**Main Review:**

**Strength**: this paper identifies an interesting phenomenon in deep neural networks and studies from various aspects.

**Weakness**: the empirical evidences are a bit misaligned with the claims, and it is unclear how this observations could be used. Please see below for details:

1. The mitosis phenomenon is described as formation of "clones" of neurons. However, the evidences only show that a (large enough) random subset of neurons could retain the prediction performance or even reconstruct the original outputs. From this observation, it is unclear if there are one-to-one correspondence of copies of neurons, or it's simply the same set of information are encoded in the subspace distributedly, i.e. a (linear) combination of a subset of neurons corresponds to another subset of neurons, but there is no neuron level correspondence. It would be great if the paper could empirically identify the neurons and their "clones" to more directly support the claim of representation mitosis. Otherwise, the current observation might not be very different from the phenomenon of parameterization redundancy of large neural networks that are reported in many previous papers that a subset of neurons or even whole layers could be removed without affecting the network performance.

2. It is unclear what is special about the last hidden layer.

3. I find it very interesting that the representation mitosis requires using state-of-the-art training setup and continued training after training error reaches zero. If the paper could dig deeper into those situations and identify what happens in different scenarios and the underlying reason for the discrepancy, then that would be a much more interesting paper.

4. The mitosis phenomenon was not reproduced on ImageNet. The suspected reason is that the network is not big enough in this case. It is understandable that the computation cost might be high to train very large models on ImageNet. However, nowadays there are many pre-trained ImageNet models available to download online, some of which (e.g. some vision transformer models) are potentially very large. Since mitosis analysis can be applied to those pre-trained models without re-training. It would be interesting to see if they are still absent in those larger models.

5. It would be great if some applications of the observations could be demonstrated. Can we use this to improve training? Or improve architecture design? Or guide neural network pruning?

-----
**After rebuttals**: Thanks the authors for posting the rebuttals, which addresses some of my concerns. I realized some of the experiments (ImageNet) cannot be relatively cheaply done and added to the paper after reading the rebuttal. However, I'm still concerned about how general this phenomenon occurs (in simpler architecture or in more complicated dataset). Furthermore, I still find the presentation (including text description and figure illustrations) of the phenomenon as "clones" of features very confusing, which the authors acknowledged are not neuron-to-neuron one-to-one mappings. As a result, I'm keeping my current rating.


**Summary Of The Paper:**

This paper identify and studies a phenomenon called "representation mitosis": as the neural network width increases above a critical threshold, the neurons in the last layer becomes "redundant" in the sense that they start to form groups of neurons where they carry identical information, and differ from each other only by a statistically independent noise.

**Summary Of The Review:**

This paper identifies a potential interesting phenomenon in deep neural networks. However, I think the empirical evidences need to be strengthened to support the claim well.

---

> ### Author Response · Authors · 2021-11-13
> **Response to reviewer 7hrx**
>
> We thank the reviewer for the time spent in reading our manuscript and for the concerns s/he raised which will help to improve our study. We will address all of them below.
>
> > *1) The mitosis phenomenon is described as formation of "clones" of neurons. However, the evidences only show that...*
>
> Indeed our evidence shows that a large enough subset of neurons can be mapped to another set of neurons with a linear transformation, with a residual that can be approximately modeled as uncorrelated Gaussian noise. Single neurons cannot be mapped into one another (this would be equivalent to a minimum chunk size for mitosis equal to 1). But our finding is not equivalent to  parametrization redundancy, which has been indeed reported several times. In particular, we do not observe that “a subset of neurons could be removed without affecting the network performance”: as shown in figure 1 and 2 the test error of the network improves approximately as $1/\sqrt n$, where $n$ is the number of neurons which are retained.
>
> > *2) It is unclear what is special about the last hidden layer*
>
> This is a good point, and we will insert a sentence discussing this issue in the amended version. The representation of all the  layers, except the last, are non-linearly mixed together in all the architectures we considered. If one removes neurons in the (l-1) layer, all the activations of the successive layer will be modified in a manner that cannot be modeled by a linear transformation.
>
> > *3) I find it very interesting that the representation mitosis requires using state-of-the-art [...]. If the paper could dig deeper into those situations...*
>
> This concern has been raised  also by other referees. We have tried to better discuss the conditions needed for mitosis in our reply to reviewer vhrZ. In short, what can not be missing is the right amount of regularization (most importantly weight decay) and a model large enough to fit its training set. Other clarifications about other concerns raised will be given in the following posts.
>
> > *4) The mitosis phenomenon was not reproduced on ImageNet. [...] Nowadays there are many pre-trained ImageNet models available to download online, some of which (e.g. some vision transformer models) are potentially very large...*
>
> This is a good suggestion, but one of the largest vision transformers, ViT-Huge (https://arxiv.org/pdf/2010.11929.pdf), has an embedding width of 1280, much smaller than  the number of features in the hidden layer of our largest ResNet50 (8192).  Moreover we would need an ensemble of such ViT-Huge models to estimate error$_\infty$ (in our paper we take an ensemble of 20 models).
>
> However we agree that networks more expressive than ResNet50 could show mitosis on ImageNet, if they are also made wider. Unfortunately, we are at present not able to validate this conclusion systematically, due to the high computational costs required to train, say, a Wide-ResNet152 on ImageNet.
>
> >*5) It would be great if some applications of the observations could be demonstrated....*
>
> We tried to apply our observations as a guideline for practical applications, but then we decided to keep the focus of the paper on the phenomenology of representation mitosis.
>
> One potential application is the following: given a fixed memory budget it is more convenient to train a wide network or split the budget over several thinner networks and use a deep ensemble average for prediction?
>
> This problem was introduced by Chirkova et al (https://arxiv.org/abs/2005.07292). They empirically found that there is a ‘critical’ width $w^*$ above which it is convenient to split the memory budget across multiple networks while below $w^*$ it is better to invest it on a single network. They left for future works principled criteria to identify $w^*$.
> In a preliminary analysis we found that the $w^*$ can be well approximated by the 'clone size'.
> If the referee feel this is appropriate, we would be happy to expand this more and add it analysis in Supp inf.
>
> We are available to discuss these or other issues related to our work even further. We hope to have solved the main concerns raised by the reviewer.

---

> > ### Author Response · Authors · 2021-11-29
> > **Assessment authors' reply**
> >
> > Dear reviewer, as the discussion period ends today we kindly ask you to assess our answers and possibly adjust the proposed mark accordingly, if you haven't done so already. We hope we provided enough clarifications to respond to your concerns, and we'll be more than happy to give further replies within today's deadline. Thank you again for the time spent in reviewing our manuscript and for your comments.

---

### Official Review · Reviewer_vhrZ · 2021-11-03

**Correctness:** 3
**Technical Novelty And Significance:** 3
**Empirical Novelty And Significance:** 2
**Recommendation:** 5
**Confidence:** 2

**Main Review:**

Strengths:
- Introduce a novel, interesting phenomenon to understand feature learning in properly trained wide neural networks
- Sufficient empirical evidences to support the claims
- Overall a well-written paper

Weaknesses:
- The conditions to achieve representation mitosis are not pinned down precisely.
- There are some heuristic claims saying that clones appear only in *well-trained* regularized networks and do not appear for the experiments on ImageNet. The more interesting question is, given (some assumptions/conditions for) a dataset, a model architecture and a training algorithm, when and how precisely can the mitosis be reached? This could be useful for choosing models and training methods to achieve the mitosis even before training the model.

Minor comments/questions:
- How does the depth and connectivity (say, sparse vs. dense networks) of the neurons in the earlier layers affect the mechanism?
- Will achieving the mitosis generally lead to better generalization? There seems to be evidences of this on page 7.
- When choosing the chunk of neurons randomly, what guides the choice for the number (or fraction) of neurons? What if we do not choose it randomly but choose it according to some deterministic procedure instead?


**Summary Of The Paper:**

This paper provides a novel explanation of benign overfitting in wide neural networks by introducing and studying a mechanism called representation mitosis. The key idea is that if the readout layer of the properly trained network is wide enough, then its neurons could split into groups (clones) that carry identical information, and differ from each other by a statistically independent noise (this mimics what happen in mitosis, thus the name for the mechanism). Moreover, above certain threshold the number of such groups increases linearly with the width, shedding light on when features are learnt. Empirical results are provided to demonstrate the mechanism.

**Summary Of The Review:**

The mechanism of representation mitosis seems to be interesting and is worth exploring. The paper provides satisfactory explanations and ample demonstrations to help understanding the mechanism. However, some of the explanations are at best heuristic in nature and would be much stronger if they could be supported by rigorous theory. I am inclined to go with a weak reject for now.

---

> ### Author Response · Authors · 2021-11-13
> **Response to reviewer vhrZ**
>
> We thank the reviewer for the careful reading of the manuscript and for the pertinent questions asked. We will try to answer to all of them below.
>
> **Weaknesses**
>
> >*"The conditions to achieve representation mitosis are not pinned down precisely."*
>
> This is a common concern raised by other reviewers. Mitosis occurs if:
>
> 1) the model achieves zero error on the training set;
> 2) the model is trained with an optimal weight decay parameter.
>
>
> >*"There are some heuristic claims saying that clones appear only in well-trained regularized networks and do not appear for the experiments on ImageNet..."*
>
> 1) means that the model can express all the potential relevant features of the data. However if the training protocol is not regularized achieving zero training error is likely associated with overfitting ‘the noise’ of the data;
>
> 2) weight decay is a standard regularizer used to avoid overfitting the data. Out of the many forms of regularization present in a typical training setting (data augmentation, weight decay, stochastic estimates of the gradient, early stopping), we found that weight decay is the main factor that produces uncorrelated representations (*"uncorrelated"* in the sense described in Sec. 3) and a decay law as $w^{-0.5}$.
>
> We will clarify these conditions better in the revised version of the manuscript.
>
> On ImageNet condition 2. is satisfied but 1. is not. ResNet50 is too ‘shallow’ to be sufficiently expressive, even when made wider. Indeed, modern SOTA architectures on ImageNet (https://arxiv.org/abs/2103.07579) are typically deeper. We believe that mitosis can occur also on ImageNet on deeper networks. At present the computational costs to train say, a Wide-ResNet152, on ImageNet are prohibitively large.
>
>
> **Minor comments**
>
> >*"How does the depth and connectivity (say, sparse vs. dense networks) of the neurons in the earlier layers affect the mechanism?"*
>
> In preliminary experiments on DenseNet100 we saw that making the network deeper reduces the size of the clones with respect to DenseNet40. At constant width, this would imply that the improved generalization with depth could be associated with the increased number of clones the network uses for its prediction. This is just a preliminary observation and it would be interesting to validate it further in future works.
>
>
> >*"Will achieving the mitosis generally lead to better generalization?"*
>
> Yes mitosis is a footprint of a model that has learned all the meaningful features from the training data and is able to generalize them effectively on unseen test data.
>
>
> >*"When choosing the chunk of neurons randomly, what guides the choice for the number (or fraction) of neurons? ..."*
>
> Choosing the neurons with a deterministic policy would not affect the results we show. All the statistics we report are averages over 100 repetitions and are shown with their error bars. The narrow amplitude of the error bars is an indication that a given statistic (i.e. chunk accuracy, $R^2$, ...) of a random chunk is very close to the average value plotted in the figures.
>
> We are available to discuss these or other issues related to our work even further. We hope to have solved the main concerns raised by the reviewer.

---

> > ### Author Response · Authors · 2021-11-29
> > **Assessment authors' reply**
> >
> > Dear reviewer,
> > as the discussion period ends today we kindly ask you to assess our answers and possibly adjust the proposed mark accordingly, if you haven't done so already. We hope we provided enough clarifications to respond to your concerns, and we'll be more than happy to give further replies within today's deadline.
> > Thank you again for the time spent in reviewing our manuscript and for your comments.

---

### Official Review · Reviewer_KFV6 · 2021-11-05

**Correctness:** 3
**Technical Novelty And Significance:** 2
**Empirical Novelty And Significance:** 3
**Recommendation:** 6
**Confidence:** 2

**Main Review:**

## Strengths:

1. The presented mitosis effect is interesting and is clearly demonstrated in certain settings.

1. Authors also show settings where the effect is clearly absent.

1. Understanding how overparameterization improves generalization remains an open research question and this finding could stimulate fruitful future research.

1. Overall the paper is well-written and figures are of good quality.

## Weaknesses:

1. IMO the main weakness of the paper is clarity,  and I hope this can be improved during rebuttal. I have found the writing to be very concise / missing details / references / mathematical derivations to be able to fully appreciate and evaluate the claims made in the paper. Specifically:

	1. What exactly is $\textrm{error}_{\infty}$? Is it an ensemble of the full network of the largest $\max W$ considered? Or is it an ensemble of networks of width $W$, where it changes along the $x$-axis of the plot? When using chunks, is an ensemble of chunks, or of full networks of width $W$, or $\max W$? Without knowing the exact definition, it's very hard to interpret a lot of plots in the paper. Further, specific ensemble sizes should be specified in the appendix.
    1. For all plots showing the $\textrm{error} - \textrm{error}_{\infty}$, I would appreciate to also see respective plots of simply error, and loss in the appendix, especially if "error infinity" depends on width W. In this case it's not clear from the plots that the error is even going down with width, since "error infinity" could increase. If "error infinity" is a single constant for all points on the plot, this is less important.
    1. Start of page 2: *"The decay rate of −1/2 in particular implies that in this regime chunks of $w_c$ neurons can be thought as statistically independent estimators of the same features of the data, differing only by a small, uncorrelated noise"*. Could you please expand on this statement, perhaps provide some mathematical details in the appendix / references? I don't understand it currently, i.e. why, and what exactly does the $-1/2$ scaling implies.
    1. End of same paragraph: *"The accuracy of these wide networks then improves with their width because the network implicitly averages over an increasing number of clones in its representations to make its prediction"*. Similarly, could you please explain more formally what is implicit averaging and why does it improve accuracy?
    1. Middle of page 4: *"This implies that a model obtained by selecting a random chunk of $w_c > w_c^∗$ neurons from a wide final representation behaves similarly to a full network of width $W = w_c$"*. per my point above, due to the ambiguous nature of "error infinity", it is not clear to me if they actually behave similarly, i.e. have comparable test error, or only comparable $\textrm{error} -\textrm{error}_{\infty}$. Again, providing plots of only test error and loss would be much appreciated, especially if "error infinity" is not a constant but changes with width $W$ or even chunk width $w_c$.
    1. End of same paragraph: *"Furthermore, a decay with rate −1/2 suggests that the final representation of the wide networks can be thought of as a collection of statistically independent estimates of a finite set of data features relevant for classification. Adding additional neurons to the chunk hence reduces their prediction error in the same way an additional measurement reduces the measurement uncertainty, leading to the −1/2 decay."* Similarly to a comment above, I would appreciate it if this statement was made mathematically explicit in the appendix or a specific reference provided.
    1. Page 5, top: *"We call a chunk of neurons a clone if it fully captures the relevant features of the data, up to some uncorrelated random noise."* Please provide a more formal definition.
    1. Page 5 below: *"The ID of the widest representations gives a lower bound on the number of coordinates required to
describe the data manifold, and hence on the neurons that a chunk needs in order to have the same
classification accuracy as the whole representation. The ID of the last hidden representation is 2 in
pMNIST, 12 in CIFAR10, 14 in CIFAR100, numbers which are much lower that the width at which a
chunk can be considered a clone."* Could you elaborate more on the dependence between ID and $w_c$? I can see why $w_c$ can't be smaller than ID, but curious if there's anything more that could be said, e.g. perhaps could be a more specific argument on how $w_c$ depends on $W$ and ID in some toy setting. It's OK if you don't have an answer, but I am curious if there are ways to ballpark estimate $w_c$ to gain better intuition into it.
	1. Top of page 6: *"In the previous paragraphs we set forth evidence in support of the
hypothesis that large chunks of the final representation of wide DNNs behave approximately like an
ensemble of independent measures of the full feature space. This allowed us to interpret the decay of
the test error of the full networks with the network width observed empirically in Fig. 2"*. As above, I don't know what an "ensemble of independent measures on the full feature space" means, or why it explains the scaling, so would appreciate much more mathematical details here.
	1. Discussion, section 4: again, I don't understand what "statistically independent learning schedule" is, and it's hard to follow the argument in the first paragraph. Would appreciate mathematical details in the appendix or a specific reference.
    1. Discussion, bottom: *"The number of clones grows linearly"* - why does it grow linearly, and not, for example, as ${W \choose w_c}$? Per above, this will also depend on what you define to be a clone exactly.
    1. Equation (1): what exactly are $\bf{x}$s? Training set activations, or validation set, or test set, or all together? Do the findings of the paper hold for all settings (train/test/train+test) above?

1. Another issue I find with the paper are the claims that mitosis explains why test error falls with width. For example, on page 9 the authors claim *"we make the observation that co-adaptation of hidden units in the form of clones occurs without
dropout, and is crucial for their improving performance with width"*; in Figure 11 the authors conclude that since the T-shaped network generalizes poorly, therefore it also can't have mitosis, which I find unjustified, and think this Figure should at least show the standard mitosis plot like in Figure 1. Overall I did not find any evidence in the paper that mitosis is responsible for improved generalization with width, if anything, Figure 8 shows that it's not. To make claims about mitosis being responsible for improved generalization with width you would need to run many more experiments, comparing generalization scaling of networks with and without mitosis. But I'm pretty sure if mitosis does not arise without data augmentation and/or weight decay as the paper claims, then it can't be responsible for improved generalization with width, since non-regularized networks still improve their generalization with width even without regularization (see e.g. https://arxiv.org/pdf/1412.6614.pdf).

1. I appreciate that you have identified that mitosis requires data augmentation and/or weight decay. It would be very interesting to ablate this further, and identify which is responsible for it exactly, especially since these are two very different techniques. In general, I wish there was more insight into why exactly the conditions you listed in the paper are necessary for mitosis.

## Minor:

1. Page 1, extra space after Sec. A.2
1. Figure 1/2 - inconsistent y-axis name w/ and w/o "log".
1. Figure 1 - no y-axis ticks.
1. Page 4, extra comma after "significantly"?
1. Figure 2: why stop so narrow for FC5, isn't this the cheapest setting, where width could be taken to be much wider than other more expensive models?
1. Page 6 bottom - "(orange)" -> "(blue)"?
1. Page 7" "done CIFAR100" -> "done on".
1. Figure 5 - would prefer matching chunk sizes and colors in (a) and (b), and plotting all panels from initialization, with shared $x$-axis.
1. Figure 6 (b) - does $x$-axis start at 0? If so, how come thick and thin lines don't start at the same $y$-values?
1. Page 8: "much higher than what observed" - "we have observed"?
1. Page 9: "found that that".
1. Page 9: "along those line" - plural "lines"?
1. Figure 9: "blu" -> "blue".


**Summary Of The Paper:**

The paper shows empirically that under certain conditions (overtraining, large width in all layers, data augmentation and regularization) neural networks (NNs) tend to learn redundant representations in the last layer. Precisely, a large enough random subset of the penultimate layer activations of size $w_c$ can almost perfectly linearly predict the whole activation of width $W$.

Further, the training error when pruning the network to these random subsets also goes down (lagging behind the training error of the full network) to 0 monotonically as training progresses.

Test error of these pruned networks appears to go down as $w_c^{-1/2}$, similarly to the test error of regular networks going down as $W^{-1/2}$.

The authors put forward some preliminary interpretation of the effect and suggest this redundancy effect could be causally responsible for the phenomenon of generalization of wide networks improving with width.


**Summary Of The Review:**

# Post-rebuttal update

The authors have clarified most of my questions, and I am raising my score to weak accept. My original conclusion remains similar, but improved clarity makes it a stronger submission. Recap:

Why accept:
* The empirical phenomenon of mitosis is interesting and shown to be robustly present in some cases, but absent in others.
* The paper is well-written and clear (given recent updates and replies to my questions, and a few more clarifications that I expect to see in the final revision).

Why weak:
* Exact conditions for mitosis are still not well understood. In the rebuttal the authors have updated the condition set to 100% training set fit + "optimal" [what is optimal?] weight decay), which is good, but I still don't think this is precise enough, and most notably the paper does not contain enough systematic experimental evidence to confirm it (e.g. mitosis measurements in wide FC networks, sweeping the weight decay from 0 to X, and observing that the rate of mitosis increases respectively).
* Exact implications of mitosis are also not well understood / demonstrated (e.g. what is the relationship between mitosis and generalization? Can a network demonstrate improved generalization with increasing width without mitosis? Can a network without improving generalization with width exhibit mitosis? etc).

# Original Review


The paper presents an interesting empirical finding, but in my opinion does not explain very well why mitosis occurs, or what are the implications of it, which is why I am leaning to reject it at this time. However, many of my concerns could be addressed in the rebuttal, so I might change my score.

---

> ### Author Response · Authors · 2021-11-15
> **Response to reviewer KFV6**
>
> We thank the reviewer for his detailed and diligent reading of our paper, from which we took a lot of cues on how to clarify the explanations in the paper. We reply to some general points raised by Reviewer KFV6 below, pointing to some specific changes, then we give shorter answers to other points raised by the reviewer.
>
>
> >**First set of concerns**
>
> >*1-2-5 What is $error_\infty$? Is the raw error of the networks decreasing with width? What is the loss?*
>
> In each plot, $error_\infty$ is the error of an ensemble of 20 instances of the widest model. In Fig. 1 for example, $error_\infty$ is the error of an ensemble of 20 DenseNet40s with width 1408. $Error_\infty$ is hence a single constant for each plot, both for the chunks and the full networks. By subtracting $error_\infty$ we followed the convention followed, for example, by Geiger et al. [J. Stat. Mech 2020] We understand now that this information was hard to gather from the plots, so we have followed the referee’s suggestion and added an additional y-axis to show the raw error in all the error plots in the paper (Fig. 1, Fig 3, Fig. 6) The training losses can be found in Fig. 4 for both the full network and the chunks.
>
> > *What is the error of a chunk?*
>
> The error of the chunks is calculated by selecting a random subset of w neurons from the last layer of the widest network (in Fig. 1, the last layer of the DenseNet has 1408 neurons). We then re-evaluate the test error using only the selected neurons, without any retraining. We repeat this procedure 100 times to obtain the error bars. We are thus **not ensembling over the chunks.**
>
> > *5 Similarity between clones and full networks*
>
> By **similar behavior of chunks**, we mean that chunks of width $w$ have the same error (and error -error$_\infty$) as a network of the same width for large enough chunks.
>
> >*7 Formal definition of a clone*
>
> A “clone” is a chunk of last hidden layer neurons that:
> 1. fit the training set and has the same test error as a network of width $w$;
> 2. It reconstructs the full final representation with a linear fit and an $R^2>0.99$.
>
> > *3-6-9 The implications of a decay rate of -½*
>
> There are several pieces of evidence that we present in the paper suggesting that the decay rate of -½, which we observe experimentally, is due to a central-limit theorem type behaviour at the level of representations. We only preview that evidence on p. 2, and we understand now that our description is too terse. We summarise our evidence here, and will clarify the writing in our manuscript accordingly.
>
> Our hypothesis is that if the network’s error decays as width^-½, this is because the network duplicates the information in the last hidden representation of the input, which is used for the final classification. The pieces of evidence supporting  this hypothesis are as follows:
> Randomly selected subsets of $w_c$ neurons from the final layer of the widest network, or *“chunks”*, achieve the same accuracy as a network of width $w_c$ (Fig. 3). Chunks beyond a minimal number of neurons allow almost complete reconstruction of the full representation using a simple linear fit (green lines, Fig. 4d-f). We call such a chunk a “clone” of the full representation (see the full definition above)
> The errors made in the reconstruction of the full representation using the linear fit are uncorrelated for two different clones. This suggests that the error in the reconstruction is due to some white noise, rather than missing some relevant features of the full representation (green lines, Fig. 4d-f).
>
> > *4. Implicit averaging over increasing number of clones*
>
> As we said above a clone achieves zero error on the training set. In these conditions a network wider than the clone size can not improve its performance on the training set. However it improves the performance on the test set, and we hypothesize that it happens because it takes an average over the test predictions made by the clones.
>
> When estimating a number, for example the class of an image, averaging over several measurements that only differ by some uncorrelated noise will lead the error to decrease as $1 / \sqrt n$, where $n$ is the number of measurements. We know from the analysis of Fig. 4d-f that large enough chunks differ only by statistically independent noise; the fact that the network error decreases as $w^{-½}$ then suggests that the network is taking an (implicit) average over them.
>
> >**Other comments**
>
> 8. We agree with the reviewer that understanding the **relationship between ID and the critical chunk size $w_c$** is an interesting direction. We are planning to investigate it using toy models. In this paper, we wanted to focus on the experimental data to permit more members of the ICLR community to study these questions.s.
>
> 10. By statistically independent learning schedule, we mean that we train the same network starting with a different random initialization on the weights.

---

> > ### Author Response · Authors · 2021-11-15
> > **Response to reviewer KFV6 - Part 2**
> >
> > 11.
> > The reviewer is correct that the number of possible ways to subsample $w_c$ neurons from a final representation of W neurons is given by a binomial factor. When we refer to the number of clones, we instead think about a concrete realisation of such a subsampling. Since the linear reconstruction shows that chunks of a certain size are statistically equivalent, we take the number of clones to be the width of the full representation divided by the size of the clone
> >
> > 12. $x^W$  are the $W$  activations randomly chosen from the full representation. We didn’t specify if these activations are taken from the training or test set because the analysis of the correlation can be done in both sets. In Fig. 4 d-e-f the analysis is done on the test activations (the same as Fig. 3 a-b-c).
> >
> > Indeed this paragraph was written to support the hypotheses that when the test error of a chunk decays with w_c^-0.5 (shaded orange area in Fig 3 a-b-c) the chunk (of the same test activations) can reconstruct the full layer with negligible error, and that the small error can be modeled by gaussian white noise.
> >
> > The analysis on the training set shows comparable results with minor differences due to the fact that the network is trained directly on these data. Namely, we find a higher R^2 coefficient and that the correlation between the residuals is smaller than on the test set. We will clarify which representation we analyze in Fig 4-e-f-g in the revised manuscript.
> >
> > > **2. Generalization**
> >
> > We do not mean to say that mitosis is causally responsible for improving generalisation in increasingly wide networks - such causal claims are very hard to show, as the reviewer correctly points out. Instead, our goal is showing that mitosis occurs and highlighting that it offers a possible mechanism by which increasingly wide networks improve their generalisation, instead of falling into the trap of overfitting.  In the amended manuscript we will clarify that we are not able to make any causal claim. We thank the author for pointing out the (great!) Neyshabur paper; one important point is that the networks they analyse are simple two-layer neural networks, which are known to undergo mitosis-like behaviour as was shown theoretically (cf. our discussion or the animations in this blog post by Francis Bach).
> >
> > > **3. Ablation**
> >
> > Mitosis occurs if:
> > the model achieves zero error on the training set;
> > the model is trained with an optimal weight decay rate.
> > We kindly ask to check our first two replies to Rev vhrZ for further information.
> >
> > >**Minor concerns**
> >
> > We thank the reviewer for pointing out all the typos. We corrected them in the new version of the manuscript.
> >
> > **Conclusion**
> > We thank the reviewer for acknowledging that our paper presents an “interesting empirical finding”. We hope that our additional explanations, and the changes we made and will make to the manuscript, alleviate your concerns regarding the clarity of our results. Should that be the case, we would appreciate it if you could raise the score. Should you have any further questions, we’ll be happy to reply.

---

> > > ### Comment · Reviewer_KFV6 · 2021-11-28
> > > **Thank you - raising my score!**
> > >
> > > Thank you for your clarifications and updates, and apologies for the late reply. I understand your results and interpretation better now, and am raising the score to weak accept.
> > >
> > > Some specific follow-ups:
> > >
> > > > We do not mean to say that **mitosis is causally responsible for improving generalisation in increasingly wide networks** - such causal claims are very hard to show, as the reviewer correctly points out. Instead, our goal is showing that mitosis occurs and highlighting that **it offers a possible mechanism by which increasingly wide networks improve their generalisation**, instead of falling into the trap of overfitting.
> > >
> > > I still think that the second boldened part essentially says that "in some cases, mitosis is causally responsible for generalization". While it does look plausible, I still find some references to this in the paper too strong, e.g. calling it "an underlying mechanism" for the bening overfitting; still having Figure 11 describe mitosis in the caption, but plotting test error instead, implying that high test error means no mitosis; calling co-adaptation "crucial for improving performance with width" in the dropout discussion.
> > >
> > > > We thank the author for pointing out the (great!) Neyshabur paper; one important point is that the networks they analyse are simple two-layer neural networks, which are known to undergo mitosis-like behaviour as was shown theoretically (cf. our discussion or the animations in this blog post by Francis Bach).
> > >
> > > Please note that more specific references would be appreciated here (link to blogpost missing; which exact section the discussion), but I think I understand your argument. Still, as a different example, Figure 2 in https://arxiv.org/abs/1901.01608 shows the FC5 network generalization improve with width, to my understanding without weight decay or data augmentation. So as far as I understand, either conditions for mitosis aren't pinpointed precisely, or mitosis is not causally responsible for generalization in this case. Therefore more systematic ablations to identify precise conditions needed for mitosis and/or identifying a more exact relationship between mitosis and generalization would make this paper much stronger.
> > >
> > >
> > > > We understand now that this information was hard to gather from the plots, so we have followed the referee’s suggestion and added an additional y-axis to show the raw error in all the error plots in the paper (Fig. 1, Fig 3, Fig. 6)
> > >
> > > Thank you for this great change! For completeness, I suggest adding the same for the rest of the plots of this kind.
> > >
> > > > The implications of a decay rate of -½
> > >
> > > Thank you for the explanation, I think I understand your intuition now. I still encourage you to go into even more details and write down a precise mathematical model of the phenomenon in the final revision, at least in the appendix.
> > >
> > > Thank you again for your detailed replies!

---

> > > > ### Author Response · Authors · 2021-11-29
> > > > **Thank you for revising your score and for your additional suggestions!**
> > > >
> > > > > *I still think that the second boldened part essentially says that "in some cases, mitosis is causally responsible for generalization"*
> > > >
> > > > We are willing to weaken further those additional sentences that seem to imply a causal connection between mitosis and generalization. Figure 11 in the appendix is just a a minor ablation. We wanted to point out that in order to see mitosis one must increase the width of all the network and not just that of the final layer. We didn't mean to use it to imply strong claims about causal relationship between mitosis and generalization. We will update the axis of this figure according to (Fig. 1, Fig 3, Fig. 6), and rephrase the caption.
> > > >
> > > > > *Please note that more specific references would be appreciated here*
> > > > We apologize for the missing link to Bach blog post. Here it is:
> > > > https://francisbach.com/gradient-descent-neural-networks-global-convergence/
> > > >
> > > > >  *Still, as a different example, Figure 2*
> > > >
> > > > We will write explicitly that the condition for mitosis are:
> > > > 1) the model achieves zero error on the training set;
> > > > 2) the model is trained with an optimal weight decay parameter.
> > > >
> > > > Generalization occurs also without mitosis but **with mitosis** it is generally **much better**.  In https://arxiv.org/abs/1901.01608 the network is trained without weight decay, and projecting the digits on the 10 principal components.
> > > > We did some initial tests on this simple setting showing that adding weight decay mitosis appears (Fig. 3 first panel), and the test error decreases from 6% to 5% (an improvement of almost 20%).
> > > >
> > > > > *For completeness, I suggest adding the same for the rest of the plots of this kind.*
> > > >
> > > > We will do it!
> > > >
> > > > Thank you again for all your detailed comments!

---

### Decision · Program_Chairs · 2022-01-20

**Decision:**

Reject

**Comment:**

This paper undertakes an empirical investigation of overparameterized neural networks, studying the last hidden representation and identifying "representation mitosis," a cloning effect whereby neurons split into groups that carry the same information. The effect is observed for a variety of architectural configurations/datasets, and a detailed set of experiments are performed to investigate the behavior.

The reviewers had split opinions about this paper, with most reviewers appreciating the novelty and salience of the observations, but with some reviewers expressing skepticism about the generality of the effect. While the experiments are thorough and revealing, the practical importance of representation mitosis remains somewhat unconvincing.

A primary motivating factor for the analysis is the search for an explanation of the unexpectedly good generalization behavior of oveparameterized networks and the origin of "benign overfitting." As highlighted in the reviews, the sensitivity of the mitosis effect to (1) training to zero loss and (2) optimal regularization suggests that it cannot be the sole explanation for benign overfitting, since the latter can and does occur without these conditions. The authors acknowledge this situation, and respond that their focus is on state-of-the-art models used by the community, rather than on toy settings. For this to be a persuasive response, more compelling results in these state-of-the-art situations should be evidenced -- in particular, as several reviewers pointed out, the negative results on ImageNet undermine this point to some extent.

Overall, representation mitosis does seem like an interesting and potentially important phenomenon, but further work is needed to develop persuasive evidence in support of the interpretations and implications. While this is a borderline submission, I believe it falls just short of the mark, and cannot recommend acceptance.